# Carbon-doped SnS$_2$ nanostructure as a high-efficiency solar fuel catalyst under visible light

Indrajit Shown [1], Satyanarayana Samireddi[1,2], Yu-Chung Chang[2,3], Raghunath Putikam[4], Po-Han Chang[5], Amr Sabbah[1], Fang-Yu Fu[2,5], Wei-Fu Chen[2], Chih-I Wu[5], Tsyr-Yan Yu[1], Po-Wen Chung[6], M.C. Lin[4], Li-Chyong Chen[2] & Kuei-Hsien Chen[1,2]

Photocatalytic formation of hydrocarbons using solar energy via artificial photosynthesis is a highly desirable renewable-energy source for replacing conventional fossil fuels. Using an L-cysteine-based hydrothermal process, here we synthesize a carbon-doped SnS$_2$ (SnS$_2$-C) metal dichalcogenide nanostructure, which exhibits a highly active and selective photocatalytic conversion of CO$_2$ to hydrocarbons under visible-light. The interstitial carbon doping induced microstrain in the SnS$_2$ lattice, resulting in different photophysical properties as compared with undoped SnS$_2$. This SnS$_2$-C photocatalyst significantly enhances the CO$_2$ reduction activity under visible light, attaining a photochemical quantum efficiency of above 0.7%. The SnS$_2$-C photocatalyst represents an important contribution towards high quantum efficiency artificial photosynthesis based on gas phase photocatalytic CO$_2$ reduction under visible light, where the in situ carbon-doped SnS$_2$ nanostructure improves the stability and the light harvesting and charge separation efficiency, and significantly enhances the photocatalytic activity.

[1] Institute of Atomic and Molecular Sciences, Academia Sinica, Taipei 10617, Taiwan. [2] Center for Condensed Matter Sciences, National Taiwan University, Taipei 10617, Taiwan. [3] Department of Materials Science and Engineering, National Taiwan University of Science and Technology, Taipei 10607, Taiwan. [4] Department of Applied Chemistry, National Chiao Tung University, Hsinchu 30010, Taiwan. [5] Graduate Institute of Photonics and Optoelectronics, National Taiwan University, Taipei 10617, Taiwan. [6] Institute of Chemistry, Academia Sinica, Taipei 11529, Taiwan. Correspondence and requests for materials should be addressed to L.-C.C. (email: chenlc@ntu.edu.tw) or to K.-H.C. (email: chenkh@pub.iams.sinica.edu.tw)

Artificial photosynthesis is one of the future energy sources that promises an environmentally friendly alternative to fossil fuels[1–4]. In this process, photocatalysts can directly harvest energy from solar light and simultaneously convert $CO_2$ to hydrocarbons, tackling both energy and global environmental problems. Photocatalytic $CO_2$ reduction to hydrocarbon fuels is a solar energy based process that requires highly efficient and stable catalytic materials[5, 6]. In the past decades, following the pioneering discovery by Inoue et al.[7] of photoelectrochemical $CO_2$ reduction in aqueous semiconductor suspensions, various semiconductor materials, in particular $TiO_2$, ZnO, NiO, $WO_3$, and $Bi_2WO_3$ have been tested as catalysts for the photocatalytic $CO_2$ reduction reaction[8, 9]. However, most of these semiconductor materials have a bandgap with energy in the ultraviolet range, resulting in low conversion efficiencies owing to their large band gaps and high charge-carrier recombination rates. To overcome these limitations and to improve photocatalytic $CO_2$ reduction efficiency, semiconductors have been modified by several strategies: nanostructuring, band gap engineering by doping and modification with metal nanoparticles, and hybridization with carbonaceous materials such as $Pt/TiO_2$, GO, g-$C_3N_4$, g-$C_3N_4$/$Bi_2WO_6$, $Cu/GO$[10–16]. Although these hybrid heterogeneous photocatalysts improved the catalytic performance significantly, the overall catalytic selectivity and quantum efficiency are far from the commercial requirements. Recently, an enzyme and semiconductor-hybrid system has been demonstrated to have a high photocatalytic $CO_2$ reduction efficiency of ~ 3–4%[17]. However, this photocatalyst system suffered from poor enzyme stability. Moreover, the most used narrow band gap semiconductor, CdS, poses another challenge due to its toxicity problem[18]. As far as photocatalytic $CO_2$ reduction is concerned, we need to develop an environmentally friendly nanostructured hybrid semiconductor, which can take us one step forward from CdS. Apart from the critical narrow band gap and high absorption coefficient to utilize maximum solar energy, efficient charge separation is the other important factor for high activity of a photocatalyst system. Therefore, controlling the carrier diffusion pathway and controlling the defects in the bulk or at the interfaces and surface of a nanostructure semiconductor are the key factors for designing a highly efficient photocatalyst system[19].

Since the discovery of graphene, two-dimensional (2D) layered transition metal dichalcogenides and metal sulfide nanostructures are playing an important role in catalysis owing to their wide range of optical and electronic properties[20, 21]. Moreover, the high surface area and low charge recombination characteristics of 2D materials can potentially enhance the photocatalyst activity[22]. Among various metal sulfides, $SnS_2$ is a naturally occurring bronze-colored n-type narrow band gap (2.2–2.4 eV) semiconductor known as mosaic gold. During the last few years, it has been demonstrated to be a promising photocatalyst for dye degradation processes[23, 24]. Recently, it has proven attractive for its potential applications as a light absorber layer for dye-sensitized solar cells[25], optoelectronics[26], gas sensing[27], and energy storage and conversion[28]. The narrow band gap with around 0.19 μm average photocarriers diffusion length[29] and the high quantum yield of $SnS_2$ thus give two advantages for a good photocatalyst under visible light. Sun et al.[30] first explored the freestanding single layer $SnS_2$ as an efficient visible-light photocatalyst for water-splitting. Recently, Sun et al.[31] have reported the photocatalytic $CO_2$ reduction to CO using $SnS_2$. However, the overall photocatalytic performance is far lower than the practical requirement due to fast recombination of the photogenerated charge carriers. To overcome this problem, it is necessary to synthesize this semiconductor nanostructure doped with metals or non-metals to control the carrier diffusion pathway and charge-carrier recombination. Semiconductor doping with non-

metal carbonaceous materials is very popular in photocatalyst systems due to their wide range of light absorption and low photo corrosion as compared with metals. These doped carbon sites act as excellent electron acceptor centers and suppress the charge recombination in the electron transfer process due to electronic interaction between doped carbon and semiconductor. Huang and co-workers[32] introduced novel carbon-doped h-BN nanosheets as a sustainable and stable visible photocatalyst system with high efficiency . In 2012, Lin et al.[33] reported enhanced photocatalytic water-splitting based on carbon-doped porous ZnO nanoarchitecture. Moreover, the most commonly reported carbon-doped photocatalysts are based on wide band gap semiconductors. Interestingly, doping can create microstrain in the crystal, which affects the electronic and optical properties. Recently, a simulation study has shown a strain-induced indirect-to direct band gap transition in bulk $SnS_2$[34]. This strain induction and non-metal doping studied in semiconductor material promises that there are possibilities of improving the photocatalytic $CO_2$ reduction activity with tuning the optoelectronic property and enhancing separation of photo induced electron-hole pairs by introducing carbon as a doping element into the semiconductor. In this work, we propose a carbon-doped $SnS_2$ nanostructure system with limited average lifetime of photogenerated electrons and holes by shortening the diffusion time so that they can reach the reaction sites before losing their energy. In the hybrid system the conductive carbon incorporated into $SnS_2$ provides the opportunities for fast charge transport in the nanostructure with an interconnected planar structure, thus shortening the diffusion time from semiconductor interior to surface reaction sites. Here, we performed the photocatalytic $CO_2$ reduction using this carbon-doped $SnS_2$ nanostructure (hereafter, referred as $SnS_2$-C) and demonstrated enhanced photocatalytic performance compared with the undoped $SnS_2$ nanoplate (hereafter referred as $SnS_2$). A theoretical study of the $CO_2$ adsorption and dissociation activity for C doped $SnS_2$ has been performed to support the experimental observation. Our results indicate that the carbon-doped nanostructure of $SnS_2$ has a key role in enhancing the visible light photocatalytic activity of the $CO_2$ reduction to solar fuels.

## Results

**Photocatalyst synthesis**. The $SnS_2$-C and $SnS_2$ photocatalyst materials were synthesized by a simple hydrothermal method, using a 1:5 stoichiometric mixture of $SnCl_4.5H_2O$ and an S source (L-cysteine or thiourea) at 180 °C. The two different sulfur sources were chosen to obtain different nanostructures resulting from their different nucleation processes and pH values during the hydrothermal synthesis. Apparently, both L-cysteine and thiourea provide a controlled condition for anisotropic growth of $SnS_2$ nanostructures and nanoplates respectively as schematically shown in Supplementary Figure 1. The detailed synthesis process is described in the Methods.

**Crystal structure analysis**. The crystal structure of the as-prepared $SnS_2$-C and $SnS_2$ were characterized by powder X-ray diffraction (PXRD) as shown in Fig. 1a. The PXRD patterns of $SnS_2$-C and $SnS_2$ match well with that of polycrystalline hexagonal $SnS_2$ berndtite (JCPDS no. 01–075–0367) and berndtite-2T (JCPDS no. 00–023–0677), respectively. The facets of $SnS_2$ show quite sharp strong intensities, which demonstrates that thiourea helps for large and thick crystal growth. The PXRD peaks of $SnS_2$-C are broader than the peaks of the $SnS_2$. The peak broadening of $SnS_2$-C implies the nanocrystal and amorphous nature of hexagonal $SnS_2$ after carbon doping. Compared with $SnS_2$, the diffraction peaks of $SnS_2$-C are slightly shifted towards

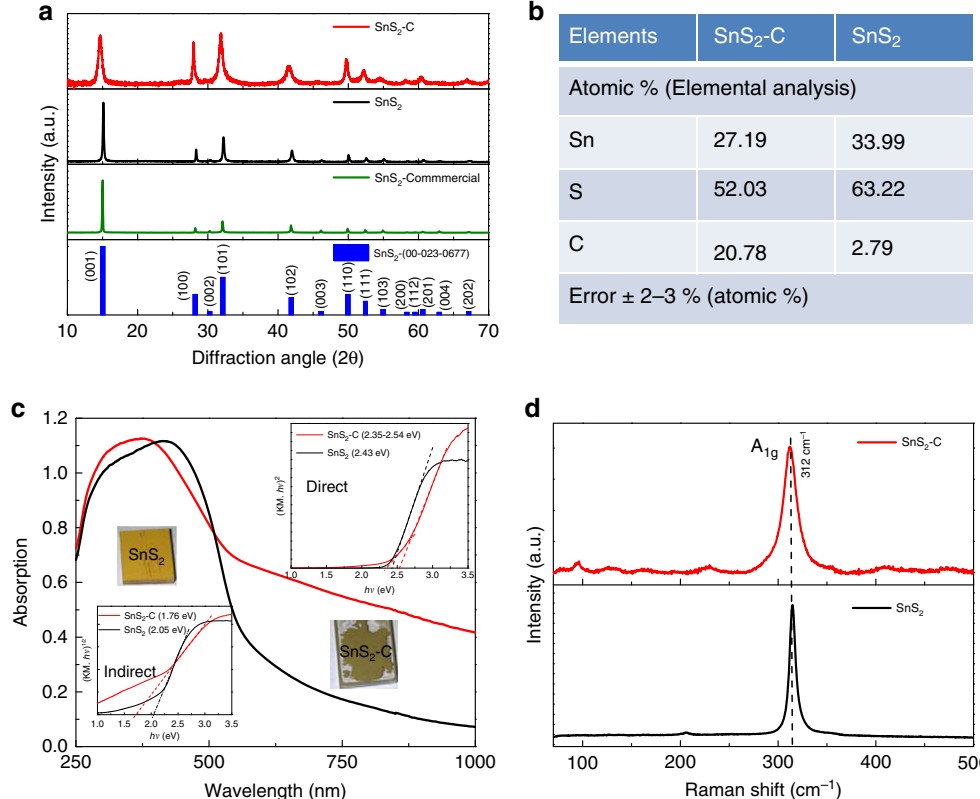

**Fig. 1** Crystal structure and optical properties of $SnS_2$-C and $SnS_2$. **a** XRD patterns of $SnS_2$-C, $SnS_2$ and commercial $SnS_2$. **b** Chemical compositions, **c** UV-vis diffuse reflectance and (insets) Tauc plots with both direct and indirect fittings, and **d** Raman spectra, of the $SnS_2$-C and $SnS_2$

lower angle. The characteristic (001) peak is quite broad and shifts from 15.12 ° to 14.66°, indicating that (001) plane growth of the $SnS_2$-C crystals is greatly inhibited and only few-layered $SnS_2$ is formed, along with lattice expansion, resulting in an enlarged d-spacing owing to the carbon doping during the hydrothermal synthesis in presence of L-cysteine. It is interesting to note that carbon doping occurred in the L-cysteine-assisted hydrothermal process, but not in the thiourea-assisted counterpart. The d-spacing of the (001) plane of $SnS_2$-C is calculated to be 0.604 nm, which is slightly larger than that of $SnS_2$ (0.585 nm). The decreased number of layers and the enlarged inter-layer spacing of $SnS_2$-C could be attributed to the structural strain generated by the expansion of the crystal lattice after interstitial incorporation of carbon. The crystallite size, microstrain and lattice d-spacing of $SnS_2$-C and $SnS_2$ are summarized in Supplementary Table 1. Interestingly, we observed that the crystallite size in $SnS_2$-C is smaller than that in $SnS_2$ and the corresponding microstrain significantly enhanced around 3.6, 3.3 and 1.8 times based on (001), (101), and (110) planes after interstitial C doping into $SnS_2$. Elemental analysis is adopted to identify the elemental compositions of the $SnS_2$-C and $SnS_2$ samples as depicted in Fig. 1b, which shows that the $SnS_2$-C sample contains around 20.78 atomic % C. Although $SnS_2$ shows ~ 2.79 atomic % of C, however, this low C level is very close to the error limit. The calculated atomic ratio of S to Sn is ~ 1.92 and 1.86 for $SnS_2$-C and $SnS_2$, respectively.

**Optical properties**. The optical absorption measurement was performed, followed by a tauc plot to estimate the band gap for the as-prepared $SnS_2$-C and $SnS_2$ (Fig. 1c). The observed direct band gaps of $SnS_2$-C and $SnS_2$ are 2.54 and 2.43 eV, respectively. In addition, $SnS_2$-C shows an absorption band edge towards

longer wavelength ~ 2.34 eV, indicating a decreased band gap in $SnS_2$-C as compared with undoped $SnS_2$. It is also worth noting that the $SnS_2$-C exhibits a significantly higher absorption ranging from a visible-light wavelength of 530 nm, which is the most intense region in the solar spectrum, towards longer wavelength. In addition, $SnS_2$-C exhibits an indirect band gap 1.75 eV that is smaller than its direct band gap. For $SnS_2$, on other hand its indirect band gap (2.05 eV) is nearly close to its direct band gap. Presumably, the interstitial C doping creating microstrain on $SnS_2$-C, as compared with $SnS_2$, affects the electronic character of the valance band and conduction band edges. This is closely similar with the recently strain-induced band gap transition on bulk $SnS_2$ simulation study[34]. Thus, we expect that the indirect band gap and additional band edge in $SnS_2$-C, whereas maximize photon absorption, will also affect the electron-hole pair's life-time, which is beneficial for charge carrier to participate in the surface photocatalytic application.

To further quantify the doped carbon content in $SnS_2$-C, we have performed solid-state NMR spectroscopic characterizations as shown in Supplementary Figure 2 and 3. The $^{13}C$ Cross-Polarization Magic Angle Spinning (CPMAS) NMR spectra of $SnS_2$-C shows two broad resonances at 23.8 ppm and at 133.3 ppm, respectively. The observed chemical shifts do not match those of pristine L-cysteine or the related decomposition compound (pyruvic acid) as shown in Supplementary Table 2. The observed resonance in the $^{13}C$ CPMAS NMR spectrum of $SnS_2$-C remains unchanged after the light activated $CO_2$ reduction reaction. Thus, we believe that the observed chemical shifts in the $^{13}C$ CPMAS NMR spectra of $SnS_2$-C should be related to the carbon containing molecules doped inside the $SnS_2$ layer during hydrothermal synthesis.

The samples were further investigated by Raman spectral analysis as shown in Fig. 1d. The spectra of both samples illustrate

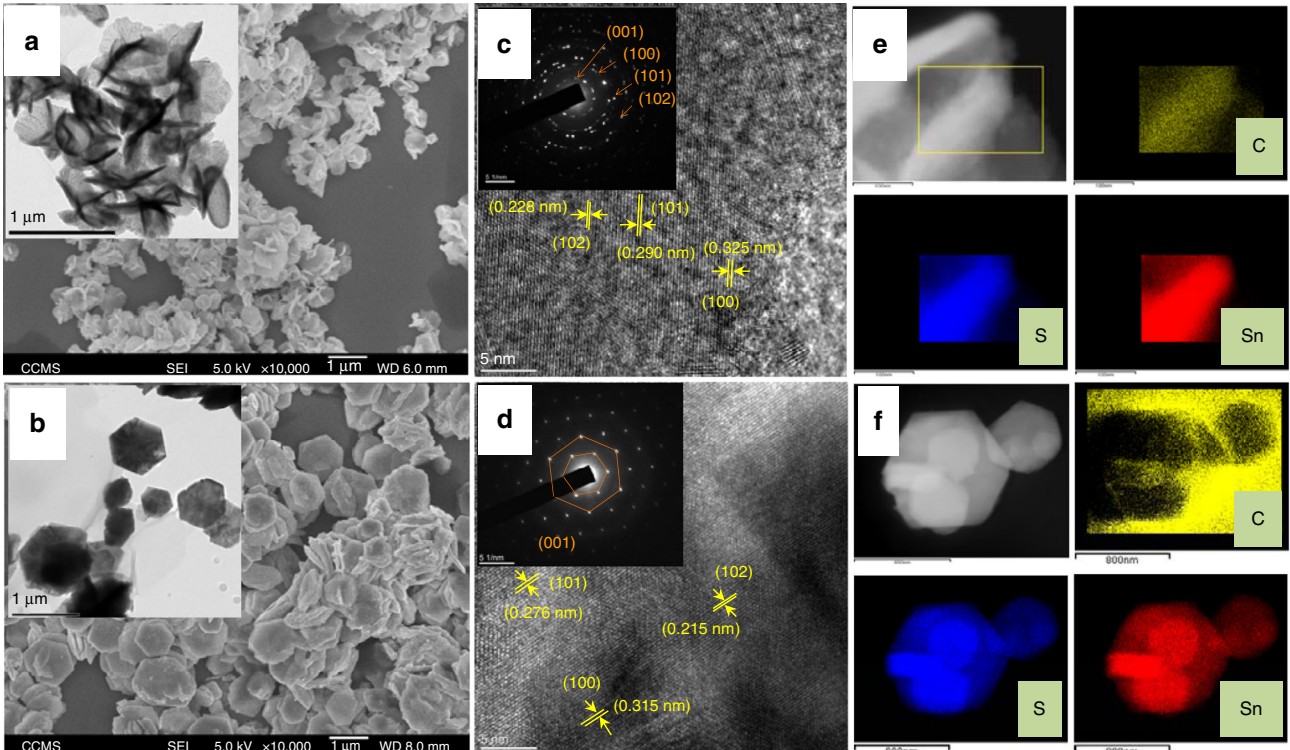

**Fig. 2** Morphology and microstructure analysis of SnS$_2$-C and SnS$_2$. **a**, **b** SEM images of SnS$_2$-C and SnS$_2$, respectively, and (inset) corresponding HRTEM images. **c**, **d** HRTEM lattice fringes and (inset) corresponding SAED patterns of SnS$_2$-C and SnS$_2$. **e**, **f** High-angle annular dark-field (HAADF) image and EDX elemental mapping of C, S, Sn from selected area for SnS$_2$-C and SnS$_2$

the strong characteristic peaks at 312 and 314.2 cm$^{-1}$, respectively, of the SnS$_2$-C and SnS$_2$ sample, which are assigned to the A$_{1g}$ mode of SnS$_2$. This observed Raman in plane mode of the atomic vibration shift around 2 cm$^{-1}$ is strongly related to the significant changes to the inter-layer covalent interaction of SnS in SnS$_2$-C after interstitial carbon doping and is well supported with the reported simulation study[34]. SnS$_2$ shows another weak peak ~ 206.1 cm$^{-1}$, resulting from E$_g$ symmetry transition owing to out of plane atomic vibration in the 2H polytype of SnS$_2$. However, for SnS$_2$-C, instead of a single E$_g$ peak in the 2H, we observed a broad peak from 190 to 225 cm$^{-1}$ corresponding to the 4H and 18R polytypes of the SnS$_2$ phase. Overall, the broadening and softening of A$_{1g}$ peak observed in the SnS$_2$-C, in comparison with SnS$_2$, can be attributed to the interstitial doping into the SnS$_2$-C layer structure and formation of different SnS$_2$ polytypes[35]. It is also worth noting that excessive incorporation of carbon may lead to formation of carbonaceous matters. As shown in Supplementary Figure 4 the Raman spectra of SnS$_2$-C reveal additional peaks ~ 1186.9, 1336.6, and 1470.1 cm$^{-1}$, which match the characteristic vibrational modes of 7A$_1$, 6A$_{1g}$, and 6E$_{2g}$, respectively, in doped amorphous carbon with pentatomic and heptatomic rings[36]. The presence of the carbonaceous matters in the SnS$_2$-C may introduce heterogeneous interfaces favorable for carrier separation, as will be discussed later.

**Morphology and microstructure analysis**. The morphology of the as-prepared SnS$_2$ samples was characterized by field emission scanning electron microscopy (FESEM). Figure 2(a, b) shows the typical SEM images of SnS$_2$-C and SnS$_2$, respectively, and their insets are the corresponding high-resolution transmission electron microscopy (HRTEM) images. The SnS$_2$-C samples exhibit flower type morphology composed of a number of nanosheets having uniform sheet dimension ~ 300–400 nm; however, these

aggregated nanosheets have rough surface. Typical thickness of the SnS$_2$-C nanosheets is ~ 30–60 nm, well underneath the photogenerated carriers diffusion length of SnS$_2$ crystal, which is more favorable for the carrier diffusion process during photocatalytic reaction. For the SnS$_2$ samples, we observed plate-like nanostructures, where the nanoplates are thicker and bigger than those nanosheets in SnS$_2$-C. The nanoplates are ~ 0.5–1 micron in size and 150–250 nm in thickness, whilst showing a smooth surface morphology. Figure 2(c, d) shows the HRTEM lattice fringes of the SnS$_2$-C and SnS$_2$, respectively, and their insets are the corresponding selective area electron diffraction (SAED) patterns. The SAED of SnS$_2$-C reveals the polycrystalline nature and dominant 001, 100, 101, and 110 diffraction planes with other planes, whereas SnS$_2$ shows single-crystal diffraction along the [001] axis. It shall be noted that the HRTEM image analysis indicates the interplaner spacing of SnS$_2$-C is larger than that of SnS$_2$. This result is in good agreement with the XRD analysis results. Figure 2e shows the high-angle annular dark field (HAADF) image and energy dispersive x-ray spectroscopy (EDX) elemental maps of the SnS$_2$-C samples, signifying that the Sn, S and C are evenly distributed within the SnS$_2$-C nanostructure. In Fig. 2f the HAADF-EDX elemental mapping of SnS$_2$ clearly shows Sn and S are well distributed without any elemental carbons. This is consistent with the previous elemental analysis data.

**Chemical composition and photoluminescence study**. Figure 3 (a, b) presents the comparison of high-resolution XPS spectra of Sn 3d and S 2p of the as-prepared SnS$_2$-C and SnS$_2$ samples. In Fig. 3a, the measured binding energies of SnS$_2$-C as compared with SnS$_2$, corresponding to Sn 3d$_{5/2}$ and Sn 3d$_{3/2}$, are higher binding energy shifted and ~ 486.7 and 495.2 eV, respectively; these binding energies indicate Sn$^{4+}$ ions in the SnS$_2$ samples. This shift is induced by the distortion of the SnS$_2$ lattice after

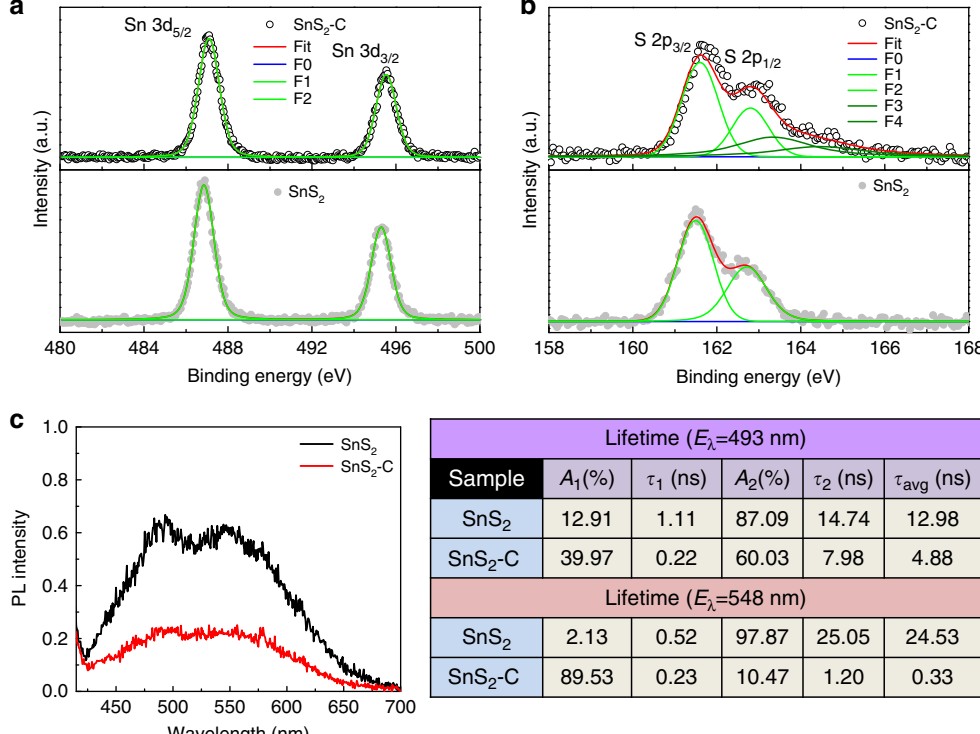

**Fig. 3** Electronic structure analysis of SnS$_2$-C and SnS$_2$. **a** High-resolution XPS Sn 3d spectra of SnS$_2$-C and SnS$_2$. **b** High-resolution XPS S 2p spectra of SnS$_2$-C and SnS$_2$. **c** Normalized PL spectra of SnS$_2$-C and SnS$_2$ and the summary table of TRPL slow, fast, and average lifetime calculated at both 493 and 548 nm emissions

carbon doping. A difference of around 8.4 eV between the two strong Sn 3d peaks is characteristic of tetravalent Sn 3d states. Furthermore, in Fig. 3b, the high-resolution S 2p core level analysis of SnS$_2$-C at binding energies of ~ 161.6 and 162.8 eV corresponds to S 2p$_{3/2}$ and S 2p$_{1/2}$, which are good typical values for a metal sulfide with a doublet separation of around 1.2 eV. The observed S 2p$_{3/2}$ and S 2p$_{1/2}$ values of SnS$_2$-C are higher binding energy shifted as compared with SnS$_2$. The observed XPS-binding energies of Sn 3d and S 2p spectra confirmed the Sn$^{4+}$ and S$^{2+}$ characters of the as-prepared SnS$_2$ samples. Interestingly, in SnS$_2$-C, we observed two extra resolvable peaks around 163.4 and 164.6 eV, which revealed the corresponding S 2p$_{3/2}$ and S 2p$_{1/2}$ states of polysulfide. The XPS results are well consistent with the reported value[37, 38]. To study the transfer and exciton separation behavior of the photogenerated electrons and holes of the as-prepared SnS$_2$ we carried out the photoluminescence (PL) and time-resolved photoluminescence (TRPL) measurements as shown in Fig. 3c (see Supplementary Fig. 5). Two PL peaks around 493 and 548 nm were observed for SnS$_2$; in contrast, these peaks become weak in case of SnS$_2$-C, revealing that the recombination of photo-induced charge carriers is reduced greatly, presumably by the enhanced interfacial charge transfer between the carbonaceous matters and SnS$_2$. The normalized PL spectrum of SnS$_2$-C shows a nearly threefold lower PL intensity as compared with that of SnS$_2$, revealing that carbon doping lowers the recombination rate. To understand the exciton separation behavior, we measured TRPL spectroscopy at 493 and 548 nm emissions to estimate the lifetime of the electron-hole pair. The emission decay data of SnS$_2$ and SnS$_2$-C were fitted biexponentially (see Supplementary Fig. 5) and the calculated slow decay time $\tau_1$, fast decay time $\tau_2$ and average lifetime $\tau_{avg}$ are summarized in Fig. 3c. The observed average lifetimes for SnS$_2$-C are 4.88 and 0.33 ns, which are much less than the 12.98 and

24.53 ns for SnS$_2$ at 493 and 548 nm, respectively. This shortening of the lifetime in SnS$_2$-C indicates the emergence of a non-radiative pathway, that is, the delocalization of electrons from SnS$_2$ to C and hence effective carrier separation. Therefore, the lower recombination of photogenerated electrons in the SnS$_2$-C allows them to reach the surface and consequently enhance the photoreduction process. Supplementary Figure 6 shows the impedance spectroscopy data for the SnS$_2$-C and SnS$_2$ on FTO electrodes at an applied potential of 1.2 V (vs NHE) with amplitude of 10 mV and a frequency ranging from 0.01 to 10$^5$ Hz in 0.1 M Na$_2$SO$_4$. The Nyquist plot reveals ideal semiconductor behavior of both SnS$_2$-C and SnS$_2$. However, the Nyquist plot for the SnS$_2$-C in the high-frequency domain shows a smaller semicircle with a low $R_C$ (270.9 $\Omega$/cm$^2$) compared with that of the SnS$_2$ (398.7 $\Omega$/cm$^2$), suggesting that the presence of carbon in the SnS$_2$-C not only improves the charge transfer behavior but also offers more conducting pathway.

**Adsorption study.** Nitrogen adsorption–desorption isotherm measurements were carried out to determine the surface area of the as-prepared SnS$_2$ samples. Supplementary Figure 7 shows the corresponding N$_2$ adsorption–desorption isotherms for both SnS$_2$ and SnS$_2$-C architectures. The shape of the curve is typical for a type II isotherm, indicating the presence of a macroporous structure for both samples. In addition, the hysteresis loops of type H3 are observed, reflecting the presence of non-rigid aggregates of plate-like particles with macropores network. Using the Brunauer–Emmett–Teller (BET) method, the specific surface area of the SnS$_2$-C was measured as about 26.56 m$^2$/g, which was larger than that of the SnS$_2$ (10.73 m$^2$/g). The two times higher surface area of SnS$_2$-C could offer more active sites exposed for CO$_2$ adsorption, thus more favorable for the photocatalytic CO$_2$ reduction to solar fuels products.

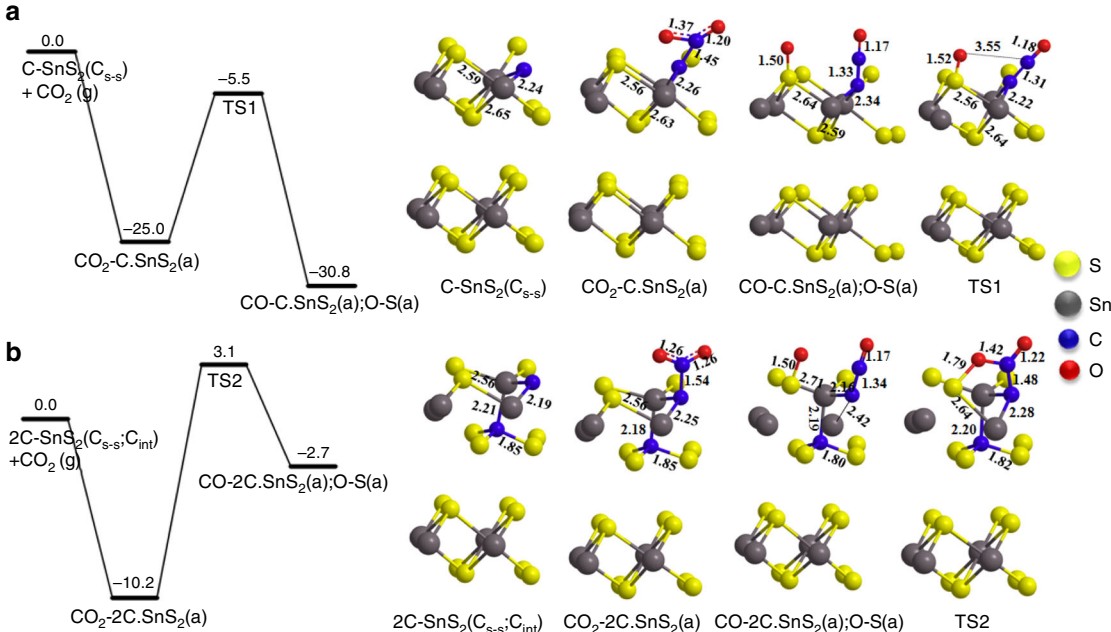

**Fig. 4** Theoretical energy calculation by DFT. **a, b** Potential comparative free energy of $CO_2$ adsorption, and dissociation energy on carbon-doped $SnS_2$[C-$SnS_2$ ($C_{s-s}$) and 2C-$SnS_2$ ($C_{s-s}$: $C_{int}$)] with their corresponding model structure (Unit: kcal mol$^{-1}$)

**Density functional theory calculation**. Photocatalytic $CO_2$ reduction activity has been shown to be primarily dependent on the adsorption energy of the $CO_2$ molecule to the photocatalyst surface and corresponding $CO_2$ dissociation energy. To understand the photocatalytic $CO_2$ reduction activity for carbon-doped $SnS_2$ surface, theoretical calculations were performed with density functional theory (DFT) plane-wave method utilizing the Vienna ab initio simulation package to predict the $CO_2$ adsorption energy and conversion into CO on the carbon-doped $SnS_2$. We considered two possible ways of introducing carbon-doping atom into the two-dimensional $SnS_2$ 2H polytype. The first was an S atom substituted with a C atom (hereafter denoted as $C_{S-S}$) (~ 12.5% atomic C doping); and the second was a C doping in an interstitial position (hereafter, denoted as $C_{int}$) (~ 25% atomic C doping). We calculated the formation energies ($E_f$) of a dopant atom in substitutional and interstitial configurations to characterize the stability of the doped $SnS_2$, as shown in Supplementary Table 3. The supercell model and partial geometries from the structurally optimized C-doped $SnS_2$ shown in Supplementary Figures 8 are explained in detail in Supplementary Method. The calculated energy results indicate that interstitial C doping is of lower formation energy than that of the S-substituted one as shown in Supplementary Table 3. We compared the energies of $CO_2$ adsorption and their dissociation energies on the two different carbon-doped $SnS_2$, C-$SnS_2$ ($C_{s-s}$) and 2C-$SnS_2$ ($C_{s-s}$: $C_{int}$), the resulting energies are presented in Fig. 4. First, $CO_2$(g) can undergo adsorption on the C doped $SnS_2$ [C-$SnS_2$($C_{s-s}$)] forming $CO_2$–C-$SnS_2$(a) with an exothermicity of 25.0 kcal mol$^{-1}$. The dissociation of $CO_2$–C-$SnS_2$(a) yielding CO–C-$SnS_2$(a) has to overcome a rather low energy barrier of 19.5 kcal mol$^{-1}$ at TS1, with an exothermicity of 30.8 kcal mol$^{-1}$. The $CO_2$ adsorption energy on the interstitial C-doped $SnS_2$ [2C-$SnS_2$($C_{s-s}$: $C_{int}$)] has a binding energy of 10.2 kcal mol$^{-1}$, which is 15 kcal mol$^{-1}$ smaller than that in the C-$SnS_2$($C_{s-s}$) case. The $CO_2$–2C-$SnS_2$(a) dissociation barrier at TS2 is only 13.3 kcal mol$^{-1}$, which is readily accessible at room temperature. In the former case, the deeper adsorption well will help accommodate more $CO_2$ than the latter case, which has, however, a lower

dissociation barrier. We therefore expect that both cases are competitive in practice. To confirm the $CO_2$ adsorption characteristic on the as-prepared photocatalyst surface, we performed the $CO_2$ adsorption study at lower temperature (195 K) as shown in Supplementary Figure 9, which could reveal more structural information of porosity instead. The total pore volume of $SnS_2$-C analyzed by $CO_2$ adsorption ($P/P_o$ = 0.96) at 195 K is 1.5-fold higher than the one of $SnS_2$ and this aforementioned comparison suggested that $SnS_2$-C might possess higher pore volume with narrow porosity (<0.4 nm) than $SnS_2$ while physically adsorbing $CO_2$. It should be mentioned that the present data is consistently comparable with the total pore volumes analyzed by nitrogen sorption ($P/P_o$ = 0.97) performed at 77 K previously, in which it was observed that the total pore volume of $SnS_2$-C is 2.1-fold higher than the corresponding one of $SnS_2$. Moreover, the overall $CO_2$ adsorption isotherm study is comparable with the theoretical prediction. Thus we believe that $SnS_2$-C can offer more active sites and different surface energy exposed for $CO_2$ adsorption.

**Photocatalytic $CO_2$ reduction study**. Photoreaction characteristics of the as-prepared $SnS_2$-C and $SnS_2$ nanostructures were determined through reaction between $CO_2$ and water in the gas phase (see Supplementary Figure 10). Figure 5a illustrates the cumulative acetaldehyde production yield after 14 h for the $SnS_2$-C and $SnS_2$ nanostructure photocatalysts. The observed maximum cumulative acetaldehyde yields after 13 h are around 125.66 µmole/100 mg$_{cat}$ and 0.55 µmole/100 mg$_{cat}$ for the $SnS_2$-C and $SnS_2$ photocatalysts, respectively. It can be seen that the prepared $SnS_2$-C nanostructure photocatalyst exhibited prominent photocatalytic $CO_2$ reduction activity under visible light and selectively produced acetaldehyde as a major product through multi-electron reduction. $SnS_2$-C photocatalyst performance started slow decay after 12 h performance as shown in Supplementary Figure 11. The maximum photocatalytic performance was observed at 9 h, after that slow decay started and showed ~ 6 % deterioration after 5 h. We believe that this is owing to the

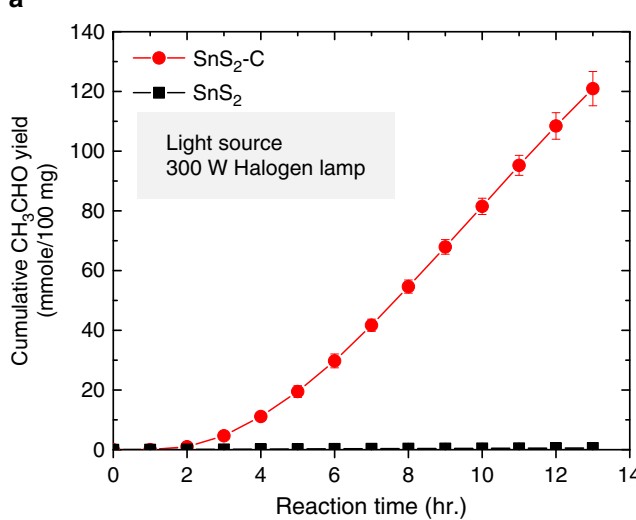

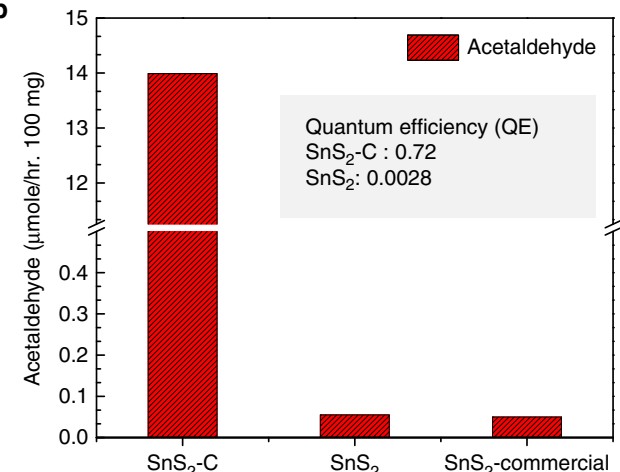

**Fig. 5** Comparative photocatalytic $CO_2$ reduction activity of $SnS_2$-C and $SnS_2$. **a** Cumulative acetaldehyde formation yield of $SnS_2$-C and $SnS_2$. **b** Comparative solar fuel formation rate and quantum efficiency of $SnS_2$-C, $SnS_2$, and commercial $SnS_2$ under a visible light source (300 W halogen lamp)

generation from $CO_2$ under visible light. In addition, to confirm the acetaldehyde formation from $CO_2$ reduction, we have conducted an isotope tracer experiment under $^{13}CO_2$ atmosphere. Analysis of the gas chromatography-mass spectrometry (GC-MS), as shown in Supplementary Figure 12, revealed the reaction product to be $^{13}CH_3^{13}CHO$ ($m/z$=46), without any product containing $^{12}C$, confirming that acetaldehyde was indeed produced from the photocatalytic reduction of $^{13}CO_2$ ($m/z$ = 45). To understand better visible-light absorption and photocatalytic $CO_2$ reduction performance of $SnS_2$-C, we performed the wavelength dependence photocatalytic reaction and summarized the PCQE in Supplementary Table 5. $SnS_2$-C showed different PCQE of 1.64, 1.04, and 0.32% at $400 \pm 25$, $500 \pm 25$, and $600 \pm 25$ nm wavelength respectively using specific band pass filters. The $^{13}C$ CPMAS NMR spectra of $SnS_2$-C before and after light irradiation also confirmed that the doped C species inside $SnS_2$-C were unaltered after the $CO_2$ reduction study under light irradiation, as shown in Supplementary Figure 3. Additional XPS analysis of $SnS_2$-C after 14 h photocatalytic performance shows that Sn 3d and S 2p peaks are unchanged (see Supplementary Fig. 13). To confirm the chemical stability of the $SnS_2$-C photocatalyst we conducted XPS analysis after photocatalytic reaction 2, 6, 10, and 14 h, respectively as shown in Supplementary Figure 14. It shall be noted that the S 2p spectra were deconvoluted into four peaks (F1, F2, F3 and F4), as depicted in Supplementary Figure 13b and Supplementary Figure 14 f-j. The XPS spectra of $SnS_2$-C show nearly unchanged in the Sn (3d) and S (2p) deconvoluted peaks as a result of irradiation. The correlation between the relative peak intensity of the deconvoluted Sn(3d) and S(2p) components, as summarized in Supplementary Table 6, shows a small change in S(2p) peaks after 2 h photocatalytic reaction as compared with the pristine sample before irradiation. Moreover, after 2 h photocatalytic reaction the S (2p) components remained unchanged and stable during photoirradiation. We believe that during the initial 2 h irradiation, chemisorptions of $CO_2$ molecule and the catalyst surface occurred, causing the catalyst surface's little change, and however, after that it remained stable.

**$CO_2$ reduction mechanism**. It is well accepted that the photocatalytic $CO_2$ reduction is a multi-electron reduction. In the initial step, direct photon absorption by $SnS_2$ generates electron-hole pairs. Specifically, carbon-doped $SnS_2$-C significantly extends the absorption band of the materials into longer wavelength range (near 530 nm and above) as compared with undoped $SnS_2$. The carbon doping also promotes the $CO_2$ molecule adsorption on the surface with a relatively small dissociation barrier, as shown in simulation studies. Moreover, carbon-doped $SnS_2$-C containing smaller nanosheets with only a few atomic layers can shorten the charge diffusion time as compared with $SnS_2$. The band edge positions of the photocatalysts directly influence the photocatalytic reduction and oxidation reactions at the catalyst surface. To understand the details of the electronic state and band energy alignment of $SnS_2$-C and $SnS_2$, we performed the ultraviolet photoemission spectroscopy (UPS) study shown in Supplementary Figure 15. The work functions of $SnS_2$-C and $SnS_2$ were calculated to be 4.4 and 4.16 eV (vs vacuum level) (see Supplementary Table 7), from which the corresponding Fermi levels can be deduced. Based on the calculated Fermi levels, conduction band and valence band maxima of $SnS_2$-C and $SnS_2$, we have drawn the electronic band diagram as shown in Fig. 6. A corresponding hypothetical photoreduction mechanism has been proposed. The electronic band diagram clearly shows that the position of the frontier orbitals of $CO_2$ with respect to the conduction band position in both $SnS_2$ and $SnS_2$-C would make

absorbed product on the catalyst surface during continuous photocatalytic reaction. To avoid this problem, instead of continuous stability study we performed 8 h consecutive cycle reaction using AM 1.5 light source. After each reaction cycle, we discontinued the reaction and cleaned the reactor together with degassing the catalyst to start a new cycle. The photocatalytic stability by consecutive cycles is shown in Supplementary Figure 11 (inset). The consecutive cycle stability results revealed that the $SnS_2$-C retained its stable catalytic performances for the $CO_2$ reduction. In addition, Fig. 5b shows a comparison of visible light photocatalytic $CO_2$ reduction to solar fuel yields of $SnS_2$-C and $SnS_2$. The maximum solar fuel formation yields for the $SnS_2$-C and $SnS_2$ photocatalysts are $\sim$ 13.98 and 0.055 µmole/100 mg$_{cat}$-hr, respectively. The photocatalytic solar fuel formation yield for $SnS_2$-C is almost 250 times higher than that for $SnS_2$. The calculated solar fuel photochemical quantum efficiency (PCQE) for $SnS_2$-C is $\sim$ 0.72% (see Supplementary Table 4). To further characterize the photocatalytic behavior a control experiment in the absence of $CO_2$ and another one without light irradiation were performed. The absence of acetaldehyde detection in the control experiments confirmed photocatalytic acetaldehyde

multi-electron reduction process feasible. However, in $SnS_2$-C, interstitially doped carbon introduced somewhat longer band tail owing to the microstrain induced new electronic state penetration into the bulk. The doped carbon helps the electrons to migrate faster to the surface of $SnS_2$-C for the reduction reaction. Besides electrons, the photogenerated holes may react with the water molecules to generate oxygen, hydrogen peroxide or hydroxide radicals. The conduction band position of $SnS_2$-C with respect to the onset reduction potential energy of $CO_2$ favors ten-electron reduction on the surface of the photocatalyst. The ten-electron reduction processes are involved in the production of acetaldehyde in our experiment. The overall reactions can be described in the following equations.

$$SnS_2 + h\nu \rightarrow SnS_2\left(e^- + h^+\right) \tag{1}$$

$$H_2O + 2h^+ \rightarrow 2H^+ + 1/2O_2 \tag{2}$$

$$H_2O + h^+ \rightarrow 1/2H_2O_2 + H^+ \tag{3}$$

$$H_2O + h^+ \rightarrow {}^{\cdot}OH + H^+ \tag{4}$$

$$S^{2-} + 2h^+ \rightarrow S \tag{5}$$

$$2CO_2 + 10H^+ + 10e^- \rightarrow CH_3CHO + 3H_2O \tag{6}$$

## Discussion

The overall photocatalytic multi-electron $CO_2$ reduction mechanism is more complex than the single electron water-splitting reaction. However, in the photocatalytic $CO_2$ reduction process, the photogenerated holes move around to the surface and cannot be excluded. The S 2p XPS analysis of $SnS_2$-C reveals that the two extra deconvoluted peaks are owing to polysulfides, which may act as scavenging agents to eliminate the photogenerated holes, resulting in more efficient separation of the photogenerated electrons and holes. Moreover, the generation of $O_2$ is suppressed, whereas the yield of acetaldehyde is enhanced. On the other hand, the excess polysulfide oxidizes to elemental sulfur via redox process and suppresses the corrosion of $SnS_2$ during the photocatalytic reaction. Overall, the enhanced photocatalytic reaction may result from combined favorable situations, including band edges tuning by induced microstrain together with high surface area, reduced photocarriers diffusion

length and improved charge separation process in carbon-doped $SnS_2$. It is worth to mention that carbon-doped $SnS_2$-C can be synthesized by a simple L-cysteine assisted hydrothermal method and is an effective way to improve the photocatalytic $CO_2$ reduction activity under visible light. Nevertheless, more studies are needed to better understand the mechanism and to further enhance the photocatalytic activity and selectivity of hydrocarbon formation.

In conclusion, carbon-doped $SnS_2$-C was successfully synthesized by an L-cysteine assisted hydrothermal process and was demonstrated to be a highly efficient photocatalyst for $CO_2$ reduction under visible light. The synthesized $SnS_2$-C photocatalyst shows selective photocatalytic $CO_2$ reduction to acetaldehyde with moderately high PCQE above 0.7%. Based on various structural analyses, the C doping is mainly incorporated as interstitials, which introduce micro strains and affect electronic band structures as well as the optical properties. Moreover, DFT calculations suggest that carbon doping also promotes the $CO_2$ molecule adsorption on the surface with a relatively small dissociation barrier in C doped $SnS_2$-C. All these factors lead to significantly enhanced photocatalytic reduction of $CO_2$. We believe that carbon doping in the narrow-band gap of dichalcogenides and other metal sulfides is a promising approach to develop high quantum efficiency photocatalysts for $CO_2$ reduction to solar fuels.

## Methods

**Synthesis**. The carbon-doped $SnS_2$ nanoflower ($SnS_2$-C) and $SnS_2$ nanoplate ($SnS_2$) were prepared by hydrothermal process. All the reagents used in the experiment were of analytical grade and used without further purification. In a typical procedure, 1 mM of tin (IV) chloride pentahydrate ($SnCl_4$, $5H_2O$) and 5 mM L-cysteine ($C_3H_7NO_2S$) were added to a 60 ml of distilled water and gradually dispersed to form a homogeneous solution by vigorous magnetic stirring for 1 h at room temperature. Finally, the resulting solution was transferred into a Teflon-lined stainless autoclave. The autoclave was sealed and heated at 180 °C for 24 h. After hydrothermal reaction, the sample was cooled to room temperature naturally. The resulting product was collected by centrifugation at 8000 rpm for 10 min and washed several times with distilled water. Finally, the collected yellow $SnS_2$-C powder was vacuum-dried at 80 °C overnight. In a similar procedure using 5 mM thiourea ($CH_4N_2S$) as an S source, $SnS_2$ nanoplate ($SnS_2$) was synthesized at 180 °C for 12 h and vacuum-dried at 80 °C. The overall $SnS_2$-C and $SnS_2$ synthesis process is schematically presented in the Supplementary Figure 1.

**Characterization**. The ultraviolet-visible absorption spectrum of powder samples was measured with a Jasco V-670 spectrophotometer using an integrated sphere. The crystal structures were determined by XRD using CuKα radiation (Bruker, D2 PHASER with XFlash). The surface morphology of all samples was characterized by FESEM (JEOL, 6700F). The HRTEM (JEOL-2100) studies with SAED and EDX were also performed to determine morphology, crystal phase and elemental compositions. The Raman spectra were measured using Jobin-Yvon LabRAM HR800 with laser source of 633 nm. X-ray photoelectron spectroscopy (XPS) analysis was performed on a theta probe ESCA VG Scientific (2002) using a monochromatic AlKα as the exciting source. The peak positions of the XPS were calibrated carefully with respect to the Au 4f peak. Finally, all the XPS spectra were deconvoluted by Voigt fitting function after a Shirley background subtraction procedure. Excitation-dependent PL measurements were performed using a spectrofluorometer (Horiba Jobin-Yvon FluoroMax-4). TRPL techniques were carried out using time-correlated single-photon counting. A pulsed laser with a wavelength of 375 nm, duration of 250 fs, and repetition frequency of 20 MHz was used as the excitation source for the steady state PL and TRPL studies. The collected PL was dispersed by a 0.75 m spectrometer and detected by the photomultiplier tube. For work function and valence band maxima measurement, UPS was performed using Perkin-Elmer phi 5400 system under vacuum with Fermi energy ($E_f$) calibration using in situ deposited gold. For UPS measurement, the samples were uniformly dispersed on the gold-coated ITO. BET surface area was determined by recording nitrogen adsorption and desorption isotherms using Micromeritics ASAP 2010 Accelerated Surface Area and Porosimetry System. The total volume was calculated from the amount of nitrogen adsorbed at $P/Po$=0.97, assuming that adsorption on the external surface was negligible in comparison to the adsorption in pores. For the microporosity study, the adsorption data were acquired at relative pressure $P/Po$ between 0 and 0.01 with little incremental dose at liquid nitrogen temperature (77 K), using a *Micromeritics 3Flex* analyzer, which was used for carbon dioxide gas ($CO_2$, purity of 99.9992%) adsorption measurements as well. All samples (50 mg each) were initially degassed at 423 K for 12 h under a $1 \times 10^{-2}$ mmHg vacuum

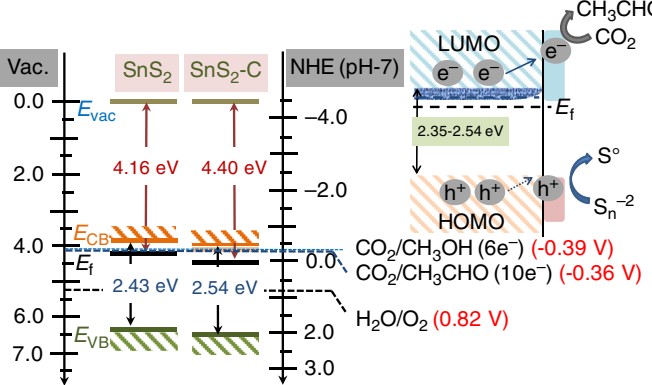

**Fig. 6** Band edge positions and photocatalytic reaction mechanism: Comparative band diagram of $SnS_2$-C and $SnS_2$, together with a proposed electron-hole separation of photo-excited electron-hole pairs in $SnS_2$-C

level, by *Micromeritics Smart VacPrep* degasser. The $CO_2$ sorption was analyzed at relative pressure ($P/P_0$) between 0.0003 and 0.96 for the $P_0$ of 789.5 mmHg at 195 K. During the analysis, the temperatures were maintained by the slurry of the combination of dry ice and acetone (wt/wt = 0.86) in Dewar. After the analysis, the free space of the sample tube was determined by using Helium gas (purity of 99.9992%). The photocatalytic $CO_2$ reduction products were analyzed by gas chromatography (GC). The GC analyses were performed on a GC-FID-CHINA CHROMATOGRAPHY 9800 system using glass column Porapak Q (80–100 mesh), at injection temperature of 50 °C, FID temperature of 150 °C and oven temperature of 80 °C. GC-MS analysis was performed on GC (HP6890)/MS(5973) system (column-Agilent J&W 122–7032 DB-WAX, inj. Temp. 250 °C and oven temperature of 35 °C utilizing $^{13}CO_2$ source (Cambridge Isotope laboratories, Inc. USA).

**Microstructural parameters analysis of the $SnS_2$-C and $SnS_2$.** The average crystallite size of the sample estimated using Scherrer's formula, i.e., $(D) = K\lambda/\beta\cos\theta$, where $K = 0.89$ is the shape factor, $\lambda$ is the X-ray wavelength of X-ray radiation, $\theta$ is Bragg's angle, and $\beta$ is the full width at half maximum of the respective peak. The microstrain is calculated using the following relation: $\varepsilon = \beta\cos\theta/4$. Supplementary Table 1 shows the average crystal size, microstarin, and lattice d-spacing calculation.

**Solid-state NMR analysis of $SnS_2$-C.** The $^{119}Sn$ MAS NMR spectrum of $SnS_2$-C and commercial $SnS_2$ (MKN-$SnS_2$-900 purchased from M K Implex Corp. Canada) are shown in Supplementary Figure 2. $SnS_2$-NS synthesized from cysteine shows a major chemical shift at −76 ppm, which is consistent with $^{119}Sn$ MAS NMR of commercial $SnS_2$. The observed line width broadening may be a result of greater distribution of crystal grain size. The additional small peak at around −605 ppm, could be attributed to different stacking of 4H and 18R polytype of $SnS_2$ as supported by our Raman spectroscopy. The $^{13}C$ CPMAS NMR spectra of $SnS_2$-C before and after $CO_2$ reduction reaction are shown in Supplementary Figure 3. For comparison, $^{13}C$ NMR chemical shift of L-cysteine and pyruvic acid are summarized in Supplementary Table 2. For other details of experimental procedures, please refer to the Supplementary Methods.

**Data availability**. The authors declare that data supporting the findings of this study are available within the paper and the supplementary information file.

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

## Acknowledgements

We thank the Ministry of Science and Technology (MOST, especially 103–2745-M-002–006-ASP, 104–2745-M-002–004-ASP), Academia Sinica, National Taiwan University, the Ministry of Education (MOE), Taiwan for financial support. Technical support from Nano-Core, the Core facilities for nanoscience and nanotechnology at Academia Sinica in Taiwan, is acknowledged.

## Author contributions

I.S. proposed and performed the synthesis, UV-Vis, XRD, Raman, experiments; S.S. performed the GC and SEM measurements, Y.-C.C. performed GC calibration and photocatalyst experimental setup; R.P. performed DFT calculations directed by M.C.L., I.S. and A.S. performed the GC-MS isotope tracer analysis, F.-Y.F. performed PL and TRPL measurement, T.-Y.Y. performed the solid-state NMR measurement; P.-H.C. performed the UPS measurements; P.-W.C. performed $CO_2$ adsorption analysis; I.S., W.-F.C., P.-W.C., C.-I.W., M.-C.L., K.-H.C. and L.-C.C. discussed and analyzed the results; I.S., K.-H.C. and L.-C.C. co-wrote the manuscript; K.-H.C. and L.-C.C. supervised the project.

## Additional information

**Competing interests:** The authors declare no competing financial interests.

