## [Peer Review File · Nature Communications]

Reviewers' comments:

Reviewer #1 (Remarks to the Author):

Photocatalytic CO₂ reduction reactions have attracted significant attention in recent years as a half cycle of artificial photosynthesis. In this work, the authors have presented carbon-doped SnS₂ nanostructure as an efficient visible light photocatalyst for CO₂ reduction. The novelty of the work is acceptable and the manuscript is also well constructed, but some key issues are not well addressed as listed below. Therefore, the current manuscript can not be considered for publication in Nature Communications.

1. The authors claimed that "Typical thickness of the SnS₂-C nanosheets is around 30-6-nm, well underneath the photo-generated carriers' diffusion length of SnS₂ crystal," How did the authors get this conclusion? What's the photo-generated carriers' diffusion length of SnS₂ crystal? The author should comment on this.
2. CO₂ adsorption measurements of SnS₂-C and SnS₂ are suggested to be conducted so as to demonstrate and compare their CO₂ adsorption abilities to promise CO₂ photoreduction reactions.
3. Carbon content in the SnS₂-S material is ca. 20 %, this is a quite high value. Did the carbon in the catalyst experience some changes after photocatalytic CO₂ reduction reactions? Therefore, the carbon source of the produced acetaldehyde should be solidly confirmed by ¹³C-labelled isotropic experiment.
4. How about the activity stability of the SnS₂-C catalyst? This is a common concern and should be evaluated.
5. Metal sulfide photocatalysts often suffer from photocorrosion, and thus some necessary characterizations of the used SnS₂ catalysts after photocatalysis are suggested to reveal the stability.
6. The used catalyst for photocatalytic tests is only 100 mg, but the authors presented the production of acetaldehyde in terms of umole/g. This is not reasonable, and the corresponding data should be revised. Actually, from the kinetic point of view, the activity is not always increasing linearly with the increase in the amount of the photocatalysts added.

Reviewer #2 (Remarks to the Author):

This manuscript reports the use of an L-cysteine-based hydrothermal process to prepare a carbon doped SnS₂ nanostructure, which exhibits quite good photocatalytic CO₂ conversion rate to produce hydrocarbons under visible-light irradiation. The manuscript is well prepared, however, there are a number of issues need to be further clarified before it can be considered for publication in this prestigious journal. I would suggest re-submission or majore revision, in more details,

1. The authors mentioned the XRD patterns of SnS₂-C and SnS₂ materials, does that mean the SnS₂-C and SnS₂ show different phase? Is that reasonable to compare these two materials and confirm the successful doping based on the crystal structure difference?
2. The carbon amount was around 20.78 atom%, how to explain such a high carbon content? And the authors also mentioned the SnS₂-C is yellow colour. What's the colour of SnS₂? Does carbon doping lead to colour change?
3. The authors used 300 W halogen lamp for the photocatalytic test. To better confirm the visible-light photocatalytic performance, a cut-off filter (e.g. >420 nm) is important. To confirm the increased visible light absorption, longer wavelength cut-off filter (e.g. >500 nm) should also be used to test the photocatalytic performance.
4. On Page 5, the authors claimed the formation of only few-layered SnS₂. How can the author know that "only few-layered SnS₂ is formed" from the XRD patterns? The authors should provide more details. Even though the (001) peak is a little broader compared to SnS₂, it is still very strong.
5. In Fig. 1d, there are 4 new weak peaks in the range of 50-250, and 2 new weak peaks in the

range of 400-500 for the SnS₂-C sample compared to SnS₂. The authors should explain more on these.

6. On Page 7, the SAED of SnS₂ shows the characteristics of single crystalline. however, the HRTEM image in Fig. 2d seems polycrystalline, which is not consistent with the SAED image. The authors should explain more on this.

7. The authors should cite the corresponding references to support the discussion of XPS spectra in Page 8.

8. C doping is important for the photocatalytic activity of CO₂ reduction in this manuscript. Thus, the authors should study more detail in C doping such as the chemical state of C in the SnS₂-C sample, the effect of different amount of C doping on the band edge position and the performance. EDS can only provide rough information about the ratio of elements. In addition, the authors should do more experiments to understand the mechanism. Photoluminescence spectra should provide useful information.

Response to Reviewers' comments:

Reviewer #1 (Remarks to the Author):

Photocatalytic CO₂ reduction reactions have attracted significant attention in recent years as a half cycle of artificial photosynthesis. In this work, the authors have presented carbon-doped SnS₂ nanostructure as an efficient visible light photocatalyst for CO₂ reduction. The novelty of the work is acceptable and the manuscript is also well constructed, but some key issues are not well addressed as listed below. Therefore, the current manuscript can not be considered for publication in Nature Communications.

1.The authors claimed that “Typical thickness of the SnS₂-C nanosheets is around 30-6-nm, well underneath the photo-generated carriers’ diffusion length of SnS₂ crystal,” How did the authors get this conclusion? What’s the photo-generated carriers’ diffusion length of SnS₂ crystal? The author should comment on this.

Answer:

We have referred (ref 29) Appl. Phys. B 68, 871–875 (1999) for the SnS₂ diffusion length. The reported diffusion length of photocarriers for SnS₂ is 0.38 μm (λ=633 nm) and 0.19 μm (λ=442 nm).

2.CO₂ adsorption measurements of SnS₂-C and SnS₂ are suggested to be conducted so as to demonstrate and compare their CO₂ adsorption abilities to promise CO₂ photoreduction reactions.

Answer:

Per the reviewer’s suggestion we have performed CO₂ adsorption isotherm study for SnS₂-C and SnS₂ at 298 K and the results are shown in Figure S9. The CO₂ adsorption isotherm curve shows the typical stepwise adsorption phenomena of nonporous or macroporous layer materials. SnS₂ with carbon doping (SnS₂-C) shows little higher CO₂ adsorption characteristics as compared with pristine SnS₂. The overall CO₂ adsorption isotherm study is comparable with the theoretical prediction.

3.Carbon content in the SnS₂-C material is ca. 20 %, this is a quite high value. Did the carbon in the catalyst experience some changes after photocatalytic CO₂ reduction reactions? Therefore, the carbon source of the produced acetaldehyde should be solidly confirmed by ¹³C-labelled isotropic experiment.

Answer:

We used ^{13}C CPMAS NMR analysis of $\text{SnS}_2\text{-C}$ before and after catalytic reaction to confirm whether carbon in the catalyst underwent some changes or not. The ^{13}C CPMAS NMR analysis of $\text{SnS}_2\text{-C}$ reveals that after catalytic reaction carbon is unchanged as shown in Figure S3.

Additionally, to confirm the acetaldehyde formation from CO_2 reduction, we have conducted an isotope tracer experiment under $^{13}\text{CO}_2$ atmosphere. Analysis of the gas chromatography-mass spectrometry (GC-MS), as shown in Figure S12, revealed the reaction product to be $^{13}\text{CH}_3^{13}\text{CHO}$ ($m/z = 46$), without any product containing ^{12}C , confirming that acetaldehyde was indeed produced from the photocatalytic reduction of $^{13}\text{CO}_2$ ($m/z = 45$).

Figure S12. Isotope tracer analysis. MS chromatograms and spectra of acetaldehyde produced by photocatalytic reduction of $^{13}\text{CO}_2$ with $\text{SnS}_2\text{-C}$. The inset shows the mass spectra of acetaldehyde generated under $^{13}\text{CO}_2$ atmosphere and $^{13}\text{CO}_2$.

To better understand the photocatalytic performance we have performed the wavelength dependence photocatalytic reaction and summarized the PCQE in Table S4. $\text{SnS}_2\text{-C}$ showed different PCQE of 1.64, 1.04 and 0.32% at 400 ± 25 , 500 ± 25 and 600 ± 25 nm wavelength respectively using corresponding band pass filters.

4.How about the activity stability of the $\text{SnS}_2\text{-C}$ catalyst? This is a common concern and should be evaluated.

Answer:

Per the reviewer's suggestion we have checked the stability of the SnS₂-C photocatalyst and provided in Figure S11a. SnS₂-C photocatalyst performance started slow decay after 12 hr performance. The maximum photocatalytic performance was observed at 9 hr, after that slow decay started and showed around 6 % deterioration after 5 hr.

5. Metal sulfide photocatalysts often suffer from photocorrosion, and thus some necessary characterizations of the used SnS₂ catalysts after photocatalysis are suggested to reveal the stability.

Answer:

Per the reviewer's suggestion to check the photocorrosion we have performed XPS and ¹³C CPMAS NMR analysis of SnS₂-C after 14hrs of CO₂ reduction and the results are provided in Figures S11b,c and S3. The XPS analysis shows Sn 3d and S 2p peaks are unchanged after photoirradiation, however, in the S 2p spectra, at higher binding energy polysulfide peaks have disappeared. The ¹³C CPMAS NMR analysis of SnS₂-C reveals that after catalytic reaction carbon is unchanged.

6. The used catalyst for photocatalytic tests is only 100 mg, but the authors presented the production of acetaldehyde in terms of umole/g. This is not reasonable, and the corresponding data should be revised. Actually, from the kinetic point of view, the activity is not always increasing linearly with the increase in the amount of the photocatalysts added.

Answer:

We agree with the reviewer's comment that the activity is not always increasing linearly with the increase in the amount of the photo-catalytic materials added. In our case, we have tried to use a fixed amount (100mg) of catalyst for photocatalytic CO₂ reduction and the gravimetric normalization of the product formation rate was established to avoid small deviation in mass from sample to sample, for a preliminary comparison among all the SnS₂ samples we studied. However, in photochemical quantum efficiency (PCQE) calculation, we are only considering the incoming photon flux and the number of electrons transferred.

Reviewer #2 (Remarks to the Author):

This manuscript reports the use of an L-cysteine-based hydrothermal process to prepare a carbon doped SnS₂ nanostructure, which exhibits quite good photocatalytic CO₂ conversion rate to produce hydrocarbons under visible-light irradiation. The manuscript is well prepared, however, there are a number of issues need to be further clarified before it can be considered for publication in this prestigious journal. I would suggest re-submission or major revision, in more details,

1. The authors mentioned the XRD patterns of SnS₂-C and SnS₂ materials, does that mean the SnS₂-C and SnS₂ show different phase? Is that reasonable to compare these two materials and confirm the successful doping based on the crystal structure difference?

Answer:

XRD can provide some useful clue for doping. The XRD peaks observed in both the doped and undoped SnS₂ can still be assigned to the same phase, however, we observed some slight shift in the 2-theta position of the peaks in the doped SnS₂ sample, with respect to its undoped counterpart. The cause of this slight shift is due to change of unit cell parameter by the incorporation of the doping C species into the SnS₂ unit cell. In addition to XRD, we further used Raman Spectroscopy and also observed some corresponding shifts in the Raman mode, which is in line with XRD, consistently suggesting interstitial C doping in SnS₂.

2. The carbon amount was around 20.78 atom%, how to explain such a high carbon content? And the authors also mentioned the SnS₂-C is yellow colour. What's the colour of SnS₂? Does carbon doping lead to colour change?

Answer:

Yes, carbon doping does lead to the color change. SnS₂ and SnS₂-C are little different yellow color samples. SnS₂ is yellow color, and SnS₂-C is pear or greenish-yellow color as shown in Figure 1c.

SnS₂

SnS₂-C

3. The authors used 300 W halogen lamp for the photocatalytic test. To better confirm the visible-light photocatalytic performance, a cut-off filter (e.g. >420 nm) is important. To confirm the increased visible light absorption, longer wavelength cut-off filter (e.g. >500 nm) should also be used to test the photocatalytic performance.

Answer:

Per the reviewer's suggestion we have performed the wavelength dependence photocatalytic reaction and summarized the PCQE in Table S4. SnS₂-C showed different PCQE of 1.64, 1.04 and 0.32% at 400 ± 25, 500 ± 25 and 600 ± 25 nm wavelength respectively using corresponding band pass filters.

4. On Page 5, the authors claimed the formation of only few-layered SnS₂. How can the author know that “only few-layered SnS₂ is formed” from the XRD patterns? The authors should provide more details. Even though the (001) peak is a little broader compared to SnS₂, it is still very strong.

Answer:

The HRTEM image (edge) clearly shows that SnS₂-C containing around 15 to 20 nm thicker layer structure. The 001 peak is little broader due to crystal size decrease as compared with undoped sample shown in table S1.

5. In Fig. 1d, there are 4 new weak peaks in the range of 50-250, and 2 new weak peaks in the range of 400-500 for the SnS₂-C sample compared to SnS₂. The authors should explain more on these.

Answer:

In SnS₂-C Raman spectra, instead of a single, E_g peak in the 2H, we observed a broad peak from 190 to 225 cm⁻¹ corresponding to the 4H and 18R polytypes of the SnS₂ phase. The observed weak peak within the range of 400-500 cm⁻¹ is presumably due to the defect state in the SnS₂-C.

6. On Page 7, the SAED of SnS₂ shows the characteristics of single crystalline. however, the HRTEM image in Fig. 2d seems polycrystalline, which is not consistent with the SAED image. The authors should explain more on this.

Answer:

SnS₂ is randomly oriented polycrystalline as shown in the HRTEM image. However, in SAED, when a small aperture enclosing only an area with single grain was used, we observed only single crystalline diffraction.

7. The authors should cite the corresponding references to support the discussion of XPS spectra in Page 8.

Answer:

Per the reviewer's suggestion we have added 2 references (37,38) to support the XPS analysis.

8. C doping is important for the photocatalytic activity of CO₂ reduction in this manuscript. Thus, the authors should study more detail in C doping such as the chemical state of C in the SnS₂-C sample, the effect of different amount of C doping on the band edge position and the performance. EDS can only provide rough information about the ratio of elements. In addition, the authors should do more experiments to understand the mechanism. Photoluminescence spectra should provide useful information.

Answer:

Per the reviewer's suggestion to understand the C doping and chemical state we used additional Raman spectroscopy and electrochemical impedance spectroscopy (EIS) studies of the SnS₂-C and SnS₂. Figure S4 shows the Raman spectra (800 to 1800 cm⁻¹) of SnS₂-C wherein additional peaks were observed around 1186.9, 1336.6 and 1470.1 cm⁻¹, which match to the characteristic vibrational modes of 7A₁, 6A_{1g}, and 6E_{2g}, respectively, in doped amorphous carbon with pentatomic and heptatomic rings³⁶. This new information is added inside the main text.

The effect of different amount of carbon doping on band edge position and performance is an on-going study that we are trying to adjust the carbon doping by a different technique. Obviously, this is out of the scope of the present manuscript.

Figure S6 shows the impedance spectroscopy data for the SnS₂-C and SnS₂ on FTO electrodes at an applied potential of 1.2 V (vs NHE) with amplitude of 10 mV and a frequency ranging from 0.01 to 105 Hz in 0.1 M Na₂SO₄. The Nyquist plot reveals ideal semiconductor behavior of the SnS₂-C and SnS₂. The Nyquist plot for the SnS₂-C in the high-frequency domain shows small semicircle with low RC (270.9 Ω/cm²) compared with that of the SnS₂ (398.7 Ω/cm²), suggesting that the presence of carbon in the SnS₂-C not only improves the charge transfer behavior but also offers more conducting pathway.

Figure S4. Raman spectra of the SnS₂-C and SnS₂ (800 to 1800 cm⁻¹ Raman shift)

Photocatalyst	R_c (Ω/cm^2)
SnS ₂	398.7
SnS ₂ -C	270.9

Figure S6. Nyquist plots of SnS₂ and SnS₂-C coated on FTO electrodes at frequencies ranging from 0.01 to 10⁵ Hz (1.2 V vs NHE).

To understand the transfer and exciton separation behavior of the photogenerated electrons and holes of the as-prepared SnS₂ we carried out the photoluminescence (PL) and time-resolved photoluminescence (TRPL) measurements as shown in Figures 3c and S5. Two PL peaks around 493 and 548 nm were observed for SnS₂; in contrast, these peaks become weak in case of SnS₂-C, revealing that the recombination of photoinduced charge carriers is inhibited greatly by interfacial charge transfer between amorphous carbon and SnS₂. The normalized PL spectrum of SnS₂-C shows a nearly 3-fold lower PL intensity as compared with that of SnS₂, revealing that carbon doping lowers the recombination rate. To understand the exciton separation behavior, we measured TRPL spectroscopy at 493 and 548 nm emissions to estimate the lifetime of the electron-hole pair. The emission decay data of SnS₂ and SnS₂-C were fitted triexponentially (Figure S5) and the calculated slow decay time τ_1 , fast decay time τ_2 and average lifetime τ_{avg} are summarized in Figure 3c. The observed average lifetimes for SnS₂-C are 4.88 and 0.33 ns respectively at 493 and 548 nm, which is

less than the corresponding values for SnS₂ (12.98 and 24.53 ns). This shortening of the lifetime in SnS₂-C indicates the emergence of a nonradiative pathway, that is, the delocalization of carriers from SnS₂ to C and hence effective carrier separation. Therefore, the lower recombination of photogenerated carriers in the carbon doped SnS₂ (SnS₂-C) allows them to reach the surface and consequently enhance the photoreduction process.

Figure 3c. Normalized PL spectra of SnS₂-C and SnS₂ and the summary table of TRPL slow, fast and average lifetime calculated at both 493 and 548 nm emissions.

Figure S5. TRPL spectra of SnS₂ and SnS₂-C at (a) 493 and (b) 548 nm emissions respectively.

Reviewers' comments:

Reviewer #1 (Remarks to the Author):

The authors did some revisions according to the previous comments. But the authors avoided to answering key concerns directly. I do not recommend this work for publication in Nature Communications.

1. After photoreaction for about 9 h, the SnS₂-C sample exhibits obvious deactivation for CO₂ reduction, that is, the stability and reusability of SnS₂-C catalyst is unsatisfactory. Besides, results of XPS characterizations further reveal that S element in SnS₂-C material indeed experiences obvious changes after photocatalysis. The authors should explore further insights into these issues.

2. The BET surface area of SnS₂-C is two times higher than that of SnS₂, and from such results the authors claimed that the SnS₂-C could offer more active sites exposed for CO₂ adsorption, thus more favorable for the photocatalytic CO₂ reduction to solar fuels products. But the results of CO₂ adsorption measurements indicate that no obvious difference in CO₂ adsorption is observed for SnS₂-C and SnS₂. How to explain these contradictory results?

3. In Figure 5a, the generation rate of acetaldehyde is low at the beginning, but after reaction for 6h, the reaction rate is enhanced obviously. Why?

4. How did the authors perform the ¹³C CO₂ isotope experiments? The details should be provided. The caption of Figure S12 is quite confusing.

5. The used catalyst for photocatalytic tests is only 100 mg, but the authors presented the production of in terms of $\mu\text{mol/g}$. This is quite not reasonable, and the corresponding data should be revised. Actually, from the kinetic point of view, the activity is not always increasing linearly with the increase in the amount of the photocatalysts added.

Reviewer #2 (Remarks to the Author):

The revision has made a number of changes to address the reviewers' comments, the quality of the work has been further improved. I think the revised version can be considered for acceptance. There is only one minor issue, the authors should discuss the possible reasons for the stability problem after some long-term testing (12hours).

Response to Reviewers' comments:

Reviewer #1 (Remarks to the Author):

The authors did some revisions according to the previous comments. But the authors avoided to answering key concerns directly. I do not recommend this work for publication in Nature Communications.

1. After photoreaction for about 9 h, the SnS₂-C sample exhibits obvious deactivation for CO₂ reduction, that is, the stability and reusability of SnS₂-C catalyst is unsatisfactory. Besides, results of XPS characterizations further reveal that S element in SnS₂-C material indeed experiences obvious changes after photocatalysis. The authors should explore further insights into these issues.

Answer:

In stability study for SnS₂-C for continuously 14 hr photocatalytic reaction we observed little deterioration after 9 hr. We believe that this is due to the absorbed product on the catalyst surface during continuous photocatalytic reaction. To avoid this problem, instead of continuous stability study we performed 8 hr consecutive cycle reaction using AM 1.5 light source. After each reaction cycle, we discontinued the reaction and cleaned the reactor together with degassing the catalyst to start a new cycle. The photocatalytic stability by consecutive cycles is shown in Figure S11a (inset). The consecutive cycle stability results revealed that the SnS₂-C retained its stable catalytic performance for the CO₂ reduction.

We have rechecked the previous XPS S(2p) spectra fitting parameter and this time we improved the fitting quality and were able to deconvolute into 4 peaks (F1, F2, F3 and F4) as initial SnS₂-C S(2p). The XPS spectra of SnS₂-C showed nearly unchanged in the Sn (3d) and

S (2p) deconvoluted peaks as a result of irradiation. Following figure shows the old fitting and new fitting data of S(2p) after photocatalytic reduction.

To further confirm the chemical stability of the SnS₂-C photocatalyst we conducted high resolution XPS analysis before and after photocatalytic reaction for 2, 6, 10 and 14 hr respectively as shown in Figure S12. The XPS spectra of SnS₂-C show nearly unchanged in the Sn (3d) and S (2p) deconvoluted peaks as a result of irradiation. The correlation between the relative peak intensity of the deconvoluted Sn(3d) and S(2p) components, as summarized in Table S4, showed a small change in S(2p) peaks after 2 hr photocatalytic reaction as compared with the pristine sample before irradiation. Moreover, after 2 hr photocatalytic reaction the S (2p) components remain unchanged and stable during photoirradiation. We believe that during the initial 2 hr irradiation chemisorptions of the CO₂ molecule occurred, causing the catalyst surface's little change, however, after that it remained stable.

Figure S12. High-resolution XPS spectra (a-e) Sn 3d and (f-j) S 2p with deconvoluted peaks of SnS₂-C before and after 2, 6, 10 and 14 hr. photocatalytic performance respectively.

Table S4 XPS spectra analysis (relative areal peak intensity) before and after photocatalytic reaction

Sn (3d)					
Peaks	0 hr	2 hr	6hr	10hr	14 hr
F1 (3d_{5/2})	61.50 %	61.38 %	62.74%	62.92	60.82 %
F2 (3d_{3/2})	38.50 %	38.62 %	37.26%	37.07	39.18 %
S (3d)					
Peaks	0 hr	2 hr	6hr	10hr	14 hr
F1	46.61 %	42.31 %	44.67 %	43.79 %	42.31 %
F2	29.16 %	28.10 %	27.10 %	28.66 %	29.87 %
F3	17.73 %	19.56 %	17.27 %	17.33 %	17.03 %
F4	06.50 %	10.03 %	10.96 %	10.21 %	11.08 %

To understand more insight morphology and chemical composition of the SnS₂-C photocatalyst during photocatalytic reaction additionally we performed the SEM and EDX analysis as shown in the following figure. We performed SEM and EDX analysis before and after photocatalytic reaction for 2, 6, 10 and 14 hr respectively. Moreover, comparison of the SEM images and EDX chemical compositions of the SnS₂-C after various photoirradiation shows the morphology and chemical compositions are unchanged. The photoirradiation time dependent XPS and SEM/EDX study clearly confirmed the stability of the SnS₂-C photocatalyst during photocatalytic CO₂ reduction.

Figure Morphology analysis (a-e) SEM images of SnS₂-C before and after 2, 6, 10 and 14 hr. photocatalytic performance respectively and (inset) corresponding high magnification SEM images.

Table EDX compositions (atomic %) of SnS₂-C photocatalyst before and after photocatalytic reactions

EDX compositions (Atomic %)					
Elements	0 hr	2 hr	6 hr	10 hr	14 hr
C	20.97 ± 2.07	22.25 ± 3.15	21.3 ± 3.31	20.66 ± 2.85	20.41 ± 2.77
S	54.82 ± 1.67	54.10 ± 1.99	54.69 ± 3.51	55.42 ± 1.51	55.49 ± 1.99
Sn	24.20 ± 0.52	23.65 ± 1.20	24.01 ± 1.79	23.92 ± 1.34	24.10 ± 0.78

2. The BET surface area of SnS₂-C is two times higher than that of SnS₂, and from such results the authors claimed that the SnS₂-C could offer more active sites exposed for CO₂ adsorption, thus more favorable for the photocatalytic CO₂ reduction to solar fuels products.

But the results of CO₂ adsorption measurements indicate that no obvious difference in CO₂ adsorption is observed for SnS₂-C and SnS₂. How to explain these contradictory results?

Answer:

We appreciated the reviewer's comment and in response to the reviewer's question, we performed the CO₂ adsorption study at lower temperature (195 K) as shown in Figure S9, which could reveal more structural information of porosity instead. The total pore volume of SnS₂-C analyzed by CO₂ adsorption ($P/P_0 = 0.96$) at 195 K is 1.5-fold higher than the one of SnS₂ and this aforementioned comparison suggested that SnS₂-C might possess higher pore volume with narrow porosity (< 0.4 nm) than SnS₂ while physically adsorbing CO₂. It should be mentioned that the present data is consistently comparable with the total pore volumes analyzed by nitrogen sorption ($P/P_0 = 0.97$) performed at 77 K previously, in which it was observed that the total pore volume of SnS₂-C is 2.1-fold higher than the corresponding one of SnS₂. Moreover, the overall CO₂ adsorption isotherm study is comparable with the theoretical prediction. Thus we believe that SnS₂-C can offer more active sites and different surface energy exposed for CO₂ adsorption.

Figure S9. CO₂ adsorption measurement at 195 K for SnS₂ and SnS₂-C respectively.

3. In Figure 5a, the generation rate of acetaldehyde is low at the beginning, but after reaction for 6h, the reaction rate is enhanced obviously. Why?

Answer:

In the photocatalytic reaction we calculated the acetaldehyde formation yield from every 1-hour interval and finally we presented it as cumulative way in Figure 5a. Initially during incubation process every hour acetaldehyde formation yield was low and slowly increased; however after 6 hr photocatalytic reaction photocatalyst surface also became completely activated and the hourly acetaldehyde formation yield reached the saturation level. The cumulative acetaldehyde formation yield data shows two different slopes at initial stage and the saturation stage of the reaction. The different hourly acetaldehyde formation in the initial and later stages in the photocatalytic reaction suggests the presence of an incubation process and a transition period around 6 hr., resulting in the enhanced reaction after 6 hr.

4. How did the authors perform the ¹³CO₂ isotope experiments? The details should be provided. The caption of Figure S12 is quite confusing.

Answer:

Per the reviewer's suggestion, we have provided detail isotope tracer analysis and revised the caption of Figure S12 (now becomes S13 in the revised version) in supporting information page 18-19. Moreover, we have revised the figure so that the mass spectra of ¹³CO₂ and acetaldehyde are shown in separate inset.

Isotope tracer analysis. MS chromatograms and spectra of acetaldehyde produced by photocatalytic reduction of ¹³CO₂ with SnS₂-C. The insets show the mass spectra of (a) ¹³CO₂ gas and (b) acetaldehyde generated under ¹³CO₂ atmosphere.

In the ^{13}C isotope experiment we performed the photocatalytic experiment in presence of $^{13}\text{CO}_2$ and after the reaction using GCMS we observed and identified the ^{13}C together in the CO_2 source and in the acetaldehyde product too. Specifically, $^{13}\text{CO}_2$ shows $[\text{CO}]^+$ and $[\text{CO}_2]^+$ peaks at m/z values of 29 and 45 and acetaldehyde exhibits corresponding characteristic peaks at m/z values of 30 and 46, 47.

5. The used catalyst for photocatalytic tests is only 100 mg, but the authors presented the production of in terms of $\mu\text{mol/g}$. This is quite not reasonable, and the corresponding data should be revised. Actually, from the kinetic point of view, the activity is not always increasing linearly with the increase in the amount of the photocatalysts added.

Answer:

In our photocatalytic study, we have tried to use a fixed amount (100mg) of catalyst for photocatalytic CO_2 reduction and the gravimetric normalization of the product formation rate was established to avoid small deviation in mass from sample to sample, for a preliminary comparison among all the SnS_2 samples we studied. However, per the reviewer's suggestion, we revised the production rate as $\mu\text{mole/hr}$. 100mg and change inside the revised manuscript.

Reviewer #2 (Remarks to the Author):

The revision has made a number of changes to address the reviewers' comments; the quality of the work has been further improved. I think the revised version can be considered for acceptance. There is only one minor issue; the authors should discuss the possible reasons for the stability problem after some long-term testing (12hours)

Answer:

In the stability study for SnS₂-C for continuously 14 hr photocatalytic reaction we observed little deterioration. We believe that this is due to the absorbed product on the catalyst surface during continuous photocatalytic reaction. To avoid this problem instead of continuous stability study we performed 8 hr consecutive cycle reaction using AM 1.5 light source. After each reaction cycle, we discontinue the reaction and clean the reactor together with degassing the catalyst to start a new cycle. The photocatalytic stability by consecutive cycles were shown in Figure S11a (inset). The consecutive cycle stability results revealed that the SnS₂-C retained its stable catalytic performances for the CO₂ reduction.

We have rechecked the previous XPS S(2p) spectra fitting parameter and this time we improved the fitting quality and were able to deconvolute into 4 peaks (F1, F2, F3 and F4) as initial SnS₂-C S(2p). The XPS spectra of SnS₂-C showed nearly unchanged in the Sn (3d) and S (2p) deconvoluted peaks as a result of irradiation. Following figure shows the old fitting and new fitting data of S(2p) after photocatalytic reduction.

To further confirm the chemical stability of the SnS₂-C photocatalyst we conducted high resolution XPS analysis before and after photocatalytic reaction for 2, 6, 10 and 14 hr respectively as shown in Figure S12. The XPS spectra of SnS₂-C show nearly unchanged in the Sn (3d) and S (2p) deconvoluted peaks as a result of irradiation. The correlation between the relative peak intensity of the deconvoluted Sn(3d) and S(2p) components, as summarized in Table S4, showed a small change in S(2p) peaks after 2 hr photocatalytic reaction as compared with the pristine sample before irradiation. Moreover, after 2 hr photocatalytic reaction the S (2p) components remain unchanged and stable during photoirradiation. We believe that during the initial 2 hr irradiation chemisorptions of the CO₂ molecule occurred, causing the catalyst surface's little change, however, after that it remained stable.

Figure S12. High-resolution XPS spectra (a-e) Sn 3d and (f-j) S 2p with deconvoluted peaks of SnS₂-C before and after 2, 6, 10 and 14 hr. photocatalytic performance respectively.

Table S4 XPS spectra analysis (relative areal peak intensity) before and after photocatalytic reaction

Sn (3d)					
Peaks	0 hr	2 hr	6hr	10hr	14 hr
F1 (3d_{5/2})	61.50 %	61.38 %	62.74%	62.92	60.82 %
F2 (3d_{3/2})	38.50 %	38.62 %	37.26%	37.07	39.18 %
S (3d)					
Peaks	0 hr	2 hr	6hr	10hr	14 hr
F1	46.61 %	42.31 %	44.67 %	43.79 %	42.31 %
F2	29.16 %	28.10 %	27.10 %	28.66 %	29.87 %
F3	17.73 %	19.56 %	17.27 %	17.33 %	17.03 %
F4	06.50 %	10.03 %	10.96 %	10.21 %	11.08 %

To understand more insight morphology and chemical composition of the SnS₂-C photocatalyst during photocatalytic reaction additionally we performed the SEM and EDX analysis as shown in the following figure. We performed SEM and EDX analysis before and after photocatalytic reaction for 2, 6, 10 and 14 hr respectively. Moreover, comparison of the SEM images and EDX chemical compositions of the SnS₂-C after various photoirradiation shows the morphology and chemical compositions are unchanged. The photoirradiation time dependent XPS and SEM/EDX study clearly confirmed the stability of the SnS₂-C photocatalyst during photocatalytic CO₂ reduction.

Figure Morphology analysis (a-e) SEM images of SnS₂-C before and after 2, 6, 10 and 14 hr. photocatalytic performance respectively and (inset) corresponding high magnification SEM images.

Table EDX compositions (atomic %) of SnS₂-C photocatalyst before and after photocatalytic reactions

EDX compositions (Atomic %)					
Elements	0 hr	2 hr	6 hr	10 hr	14 hr
C	20.97 ± 2.07	22.25 ± 3.15	21.3 ± 3.31	20.66 ± 2.85	20.41 ± 2.77
S	54.82 ± 1.67	54.10 ± 1.99	54.69 ± 3.51	55.42 ± 1.51	55.49 ± 1.99
Sn	24.20 ± 0.52	23.65 ± 1.20	24.01 ± 1.79	23.92 ± 1.34	24.10 ± 0.78

REVIEWERS' COMMENTS:

Reviewer #1 (Remarks to the Author):

The authors have addressed the reviewers' comments carefully; the quality of the work has been further improved. The revised version is supported for publication